# Genetic and Molecular Interactions between *H*^Δ*CT*^, a Novel Allele of the Notch Antagonist Hairless, and the Histone Chaperone Asf1 in *Drosophila melanogaster*

**DOI:** 10.3390/genes14010205

**Published:** 2023-01-13

**Authors:** Dieter Maier, Milena Bauer, Mike Boger, Anna Sanchez Jimenez, Zhenyu Yuan, Johannes Fechner, Janika Scharpf, Rhett A. Kovall, Anette Preiss, Anja C. Nagel

**Affiliations:** 1Institute of Biology, Genetics Department 190g, University of Hohenheim, Garbenstr. 30, D-70599 Stuttgart, Germany; 2Biozentrum, University of Basel, Spitalstrasse 41, CH-4056 Basel, Switzerland; 3European Center for Angioscience, Medical Faculty Mannheim, Heidelberg University, Ludolf-Krehl-Straße 13–17, D-68167 Mannheim, Germany; 4Department of Molecular Genetics, Biochemistry and Microbiology, University of Cincinnati College of Medicine, Medical Sciences Building 2201, Albert Sabin Way, Cincinnati, OH 45267, USA; 5Institute of Biomedical Genetics (IBMG), University of Stuttgart, Allmandring 31, D-70569 Stuttgart, Germany

**Keywords:** Asf1, co-repressor, chromatin dynamics, *Drosophila melanogaster*, Hairless, Notch signaling pathway, protein complex, gene regulation

## Abstract

Cellular differentiation relies on the highly conserved Notch signaling pathway. Notch activity induces gene expression changes that are highly sensitive to chromatin landscape. We address Notch gene regulation using *Drosophila* as a model, focusing on the genetic and molecular interactions between the Notch antagonist Hairless and the histone chaperone Asf1. Earlier work implied that Asf1 promotes the silencing of Notch target genes via Hairless (H). Here, we generate a novel *H*^Δ*CT*^ allele by genome engineering. Phenotypically, *H*^Δ*CT*^ behaves as a Hairless gain of function allele in several developmental contexts, indicating that the conserved CT domain of H has an attenuator role under native biological contexts. Using several independent methods to assay protein–protein interactions, we define the sequences of the CT domain that are involved in Hairless–Asf1 binding. Based on previous models, where Asf1 promotes Notch repression via Hairless, a loss of Asf1 binding should reduce Hairless repressive activity. However, tissue-specific Asf1 overexpression phenotypes are increased, not rescued, in the *H*^Δ*CT*^ background. Counterintuitively, Hairless protein binding mitigates the repressive activity of Asf1 in the context of eye development. These findings highlight the complex connections of Notch repressors and chromatin modulators during Notch target-gene regulation and open the avenue for further investigations.

## 1. Introduction

The highly conserved Notch signaling pathway mediates cell–cell communication in the process of cellular specification during development and in the context of tissue homeostasis (reviewed in Refs. [1,2]). Given its impact on human health, a comprehensive understanding of its regulation is of critical importance. To this end, model systems, such as *Drosophila melanogaster*, play a pivotal role in defining the complex interrelations of the many factors involved in signaling. Notch signaling occurs between two neighboring cells—one cell presenting a membrane bound ligand to the other cell presenting the Notch receptor. As a consequence of receptor–ligand binding, Notch target genes are activated in the signal receiving cell, directing a specific cell fate or provoking cellular change (reviewed in Refs [2,3,4]). Central to Notch signal transduction lies the DNA binding transcription factor CSL (acronym of mammalian CBF1/RBPJ, fly Su(H) and worm Lag1 orthologs), which assembles the activator or repressor complexes on Notch target enhancers and gene promoters in response to the activation or inactivation of the Notch receptor (reviewed in Refs [4,5]). The canonical view is that Notch receptor activation causes the release of the Notch intracellular domain (NICD) from the cell membrane, which localizes to the nucleus and thereby becomes a coactivator of CSL itself to activate the target genes in conjunction with other transcriptional partners. In the absence of signaling activity, CSL recruits co-repressors to target genes, resulting in gene silencing (reviewed in Refs. [6,7,8,9,10]). In *Drosophila*, the CSL homolog Suppressor of Hairless (Su(H)) binds to the Notch antagonist Hairless (H), which assembles a repressor complex together with the general co-repressors Groucho (Gro) and C-terminal binding protein (CtBP), which provide repressive histone deacetylase activity [7,11,12,13,14,15,16].

H is a large, ~1000 residue, highly basic protein found in arthropods [7,17,18]. It assembles a high-affinity 1:1 protein complex with Su(H), whose crystal structure has been solved previously [19,20]. H binding involves a short, highly conserved NT-domain, which assumes a β-hairpin motif that inserts itself into the hydrophobic core of the C-terminal domain (CTD) of Su(H). This highly unusual hydrophobic interaction causes a distortion of the β-sandwich structure of the CTD, precluding the binding of NICD [20,21]. Accordingly, the activator and repressor complex formation in *Drosophila* involves one of two allosteric states of Su(H). There is little structural information on H apart from the defined binding sites for Su(H), Gro and CtBP, as well as nuclear import and export signal sequences [7,11,12,13,14,22]. Even considering the substantially smaller H homologs from other arthropods, such as *Apis mellifera* or *Daphnia pulex*, these sites contribute less than fifteen percent of the total H protein [23,24]. This raises the question of the functional significance of other regions of H, notably the well-conserved CT domain, which is adjacent to the Su(H)-binding domain NT [19,23]. Additional reported binding partners of H include Cyclin G, Regulatory particle triple-A ATPase 2 (Rpt2), Runt and Anti-silencing factor 1 (Asf1) [25,26,27,28]. Whereas the former three proteins all contact the C-terminal half of the H protein, the binding interface between H and Asf1 remains an open question to date. Moreover, the individual roles of these H-binding partners in the regulation of Notch signaling activity have remained largely unexplored. These factors have been implicated in diverse roles, such as the specification of cell types, the participation in DNA damage repair and cell growth, the stability of signaling components and influencing the chromatin landscape [28,29,30,31,32,33].

Counterintuitively, Su(H)-H repressor complexes do not appear to stably bind DNA in the absence of Notch signaling activity but are detected there only transiently [34,35,36,37,38]. Su(H) binds very dynamically to Notch promoter DNA sites under Notch-OFF signaling conditions, whereby its residence time on DNA is increased under Notch-ON conditions [38]. Chromatin accessibility is increased under Notch-ON conditions, allowing assisted recruitment of further activators, as well as Su(H)-H repressor complexes, thereby facilitating subsequent target gene repression [38,39]. Interestingly, the genomic regions bound by H display an active chromatin status, despite its role as a repressor and similar to its co-repressor Gro [37,40]. Together with its many interaction partners, H may modulate rather than silence chromatin, thereby ensuring appropriate levels and/or timing of target gene regulation [37]. Certainly, the chromatin landscape is a critical determinant for Notch activity, which has proven be to be highly sensitive to changes in cellular signaling (reviewed in Refs. [9,10,39,41]). Accordingly, several chromatin modifiers have been linked to the regulation of Notch activity, including the histone chaperone Asf1 [28,33]. Given that Su(H) binding to DNA requires an open and active chromatin landscape, any changes to this chromatin environment could potentially affect Notch transcriptional responses.

Asf1 functions as a histone chaperone, which is important for chromatin dynamics. It guides nucleosome assembly and disassembly during replication, the DNA damage response and chromatin silencing. Work in yeast uncovered Asf1 function in transcriptional regulation, facilitating nucleosome dissociation during active transcription and H3/H4 assembly during transcriptional repression under specific cellular conditions (reviewed in Ref. [42]). In *Drosophila*, Asf1 was found to contribute to the selective silencing of Notch target genes via the general Notch antagonist Hairless (H) [28]. In this context, Asf1 is part of a large protein complex, including several different histone modifiers [33].

Here, our work demonstrates that the Asf1–H relationship is much more complex than previously appreciated. An *H* allele (*H*^Δ*CT*^) lacking the Asf1 binding site behaves—contrary to expectations—as a stronger, not weaker, inhibitor of Notch signaling, suggesting that, under native conditions, H attenuates Asf1 repressive activity during Notch signal regulation. This finding is interesting in the context of the exquisite sensitivity of Notch signaling activity and changes in the chromatin landscape, as both Asf1 and H have been implicated in the modulation of local chromatin structure. As our work fits within the general framing of Asf1 function, it may open the avenue for investigating the complex connections of Notch repressors and chromatin modulators during Notch target-gene regulation.

## 2. Materials and Methods

### 2.1. Genome Engineering of the H^ΔCT^ Allele

The generation of the *H*^Δ*CT*^ allele followed the principles outlined before [43]. Genomic full-length Hairless DNA in pBT vector (5.1 kb *Kpn* I/*Eco* RV fragment) [43] was *Xba* I/*Sac* I digested and ligated with an annealed primer pair containing a *Kpn* I recognition sequence and compatible *Xba* I/*Sac* I sticky ends. The *Eco* RI restriction site was deleted from the polylinker by a *Sma* I/*Eco* RV digest followed by re-ligation. Afterward, the CT domain coding sequences were removed together with the separating intron by *Eco* RI/*Bsp* EI digestion. An annealed primer pair carrying compatible sticky ends for *Eco* RI and *Bsp* EI was ligated to circulate the clone; it included some sequences eliminated before (see Figure 1a,c): HdCt up 5′ AAT TCT ACG CTT GGC CGA GCT GCT CT 3′ and HdCt lo 5′ CCG GAG AGC AGC TCG GCC AAG CGT AG 3′. The third intron was PCR amplified; the lower primer contained an *Eco* RI site and splice sites: Hint-dCT up 5′ CGT CGG CGG TGA CCA CAG CGT AT 3′ and HdCT lo 5′ GAG CTG TTG GAA TTC GAA ACT GCA ATG AAG AG 3′. The amplicon was reintroduced by *Eco* RI digestion and ligation. The final construct was shuttled into the pGE-attB^GMR^ transformation vector [44] via the *Kpn* I sites and sequence verified. The *H*^Δ*CT*^ allele was inserted into the *H^attP^* founder genome as described earlier [43,44]. Successful transgenic lines were confirmed by PCR. Subsequently, the pGE-attB^GMR^ and *white^+^* marker sequences were eliminated via Cre-mediated recombination [43,44]; the final line was PCR genotyped.

### 2.2. Tissue-Specific Expression Using the Gal4/UAS-System

Tissue-specific expression was performed using the Gal4/UAS system [45]. In order to avoid position effects and allow a direct comparison, the H-overexpression constructs were all placed at the identical chromosomal position at 3 L 68E via PhiC31 integrase-based insertion using the ΦX-68E strain as outlined earlier [19,46]. To this end, the deletion ΔCT (codons 271–320) was introduced in a *Cla* I fragment of the Hairless cDNA by using the ExSite PCR Based Site-Directed Mutagenesis Kit (Stratagene, La Jolla, CA, USA). The truncated *Cla* I fragment was reinserted into the full-length Hairless cDNA and cloned into pUAST-attB-vector [46] as *Acc* 65I/*Xba* I fragment; it was sequence verified. The successful transformant UAS-attB-HΔCT line was confirmed by PCR genotyping. A full-length UAS-attB-H transgenic line, likewise inserted in 68E, served as control [19]. Further UAS lines used were UAS Asf1/CyO (kind gift of S. Bray) [47], UAS-CtBP (kind gift of S. Parkhurst) [48], UAS-Gro (kind gift of C. Delidakis) [49], UAS-GFP (BL4775) [50], UAS dsAsf1 (VDRC 23737) [51], UAS dsCtBP [14], UAS dsGro [52], UAS dsH [14]. The driver lines were ey-Gal4 (BL5534) [53], gmr-Gal4 (BL1104) [54], Eq1-Gal4 (kind gift of H. Sun) [55] and omb-Gal4 [56]. Moreover, the combined lines ey-Gal4 UAS Asf1/CyO (gift of S. Bray) [47] and ey-Gal4 UAS dsAsf1/CyO were used; the latter was generated by meiotic recombination followed by PCR-based genotyping. For the ectopic expression of ey-Gal4 UAS Asf1 in the homozygous *H*^Δ*CT*^ background, we crossed *y^1^ w^67c23^*; *H*^Δ*CT*^/*H*^Δ*CT*^ with ey-Gal4 UAS Asf1/CyO flies. Red-eyed offspring carries one copy of ey-Gal4 UAS Asf1 on the second and one allele of *H*^Δ*CT*^ on the third chromosome. To avoid recombination, we crossed the respective males back to homozygous virgin *y^1^ w^67c23^*; *H*^Δ*CT*^/*H*^Δ*CT*^ females and selected again for red-eyed offspring, half of which were expected to be homozygous for *H*^Δ*CT*^ in the background of ey-Gal4 UAS Asf1. Flies were categorized by phenotype and subsequently genotyped by PCR.

### 2.3. Phenotypic Analyses

#### 2.3.1. Adult Phenotypes

Adult flies of the respective genotype were collected and etherized. Pictures were taken with an ES120 camera (Optronics, Goleta, CA, USA) mounted to a Wild M5 stereomicroscope (Leica, Wetzlar, Germany). Dehydrated wings of female flies were embedded in Euparal (Carl Roth, Karlsruhe, Germany) and documented with an ES120 camera (Optronics, Goleta, CA, USA) coupled to a Zeiss Axiophot (Carl Zeiss, Jena, Germany). Pictures were recorded using Pixera viewfinder software version 2.0. Alternatively, uncoated etherized flies were pictured with a table top scanning electron microscope NeoScope (JCM-5000 SEM; Nikon, Tokyo, Japan) using proprietary software. The wing or eye area was determined using the freehand tool of Image J (open source). Only the SEM pictures of female flies were used for the quantification of micro- and macrochaetae, respectively. The numbers of bristles were determined as described before [57,58,59,60]. Statistical analysis was conducted by ANOVA two-tailed test for multiple comparisons using Tukey–Kramer’s or Dunnett’s approach, as indicated: *** *p* ≤ 0.001 highly significant; ** *p* ≤ 0.01 very significant; * *p* ≤ 0.05 significant; not significant (n.s.) *p* > 0.05. The figures were assembled using Corel *Photo Paint, *Corel Draw** and *BoxPlotR* software.

#### 2.3.2. Clonal Analysis

The FLP/FRT system [61] was applied for generating twin clones as described [43,62,63] by crossing the FRT82B *H*^Δ*CT*^ fly line with *y*^1^
*w** hs-flp; FRT82B Ubi-GFP^S65T^nls/TM6B (BL32655, obtained from BDSC, Bloomington, IN, USA); the wild-type allele hence expressed GFP. Clones were induced by a 1 h heat shock at 37 °C in first to second instar larvae. Wing imaginal discs were dissected from wandering third instar and subjected to antibody staining according to standard procedures [62,63] using primary rat anti-Deadpan antibodies (Dpn, 1:100; ab19573 Abcam, Cambridge, UK) and goat secondary antibodies coupled to Cy3 (Jackson Immuno-Research via Dianova, Hamburg, Germany). GFP signals were recorded without further enhancement. Fluorescently labeled tissue was embedded in Vectashield (Vector labs, Eching, Germany) and pictures taken with a BioRad MRC1024 confocal microscope coupled to a Zeiss Axioskop using LaserSharp 2000^TM^ software (Carl Zeiss, Jena, Germany). The figures were assembled using *Image J,* Corel *Photo Paint* and *Corel Draw* software.

### 2.4. Yeast Two-Hybrid Experiments

Yeast two-hybrid experiments were performed as previously described using standard protocols [63,64,65,66,67,68]. As bait, we used full-length pEG H [14] and the deletion constructs pEG NTCT, pEG NT and pEG CT [23], pEG NTCT^Δ^NT and pEG NTCT^Δ^CT [19]. pJG Su(H) [14] and pJG Asf1, which was amplified from genomic DNA and inserted as *Eco* RI/*Xho* I fragment into pJG4-5 [66], served as prey. To this end, the following primers were used: Asf1-up 5′ GCA ATC TCC GAG GTG AAT TCA TGG 3′ and Asf1-lo 5′ CGG ACT GCC TCG AGC TCT CAA CA 3′. Empty vectors pEG202 [67] and pJG4-5 [66] served as control. The expression of the lacZ reporter from pSH18-34 resulted from productive interaction [68].

### 2.5. Protein Co-Precipitation Using the Myc-Trap System

#### 2.5.1. Cloning of Various NTCTmyc Constructs and of Asf1-HA in pMAL Vector

The NTCT region (Ser 212—Ile 369) was PCR amplified with the upper primer 5′ AAC TCC ***CTC GAG*** AGC TTT GAT ATG GGC AG G 3′ containing a ***Xho* I** restriction site and lower primer 5′ GAT TGC CG***A AGC TT***A AAT GGG CTG CTG ATC 3′ containing a ***Hind* III** restriction site. After a *Xho* I/*Hind* III digest, the PCR product was cloned into pBT (Stratagene, La Jolla, CA, USA), where the *Eco* RI had been eliminated beforehand. From there, the product was inserted via *Xho* I/*Hind* III sites into the pESC-LEU plasmid (Stratagene, La Jolla, CA, USA), providing an N-terminal myc-tag. Then, NTCTmyc was released as *Sal* I/*Hind* III fragment and ligated into pMAL vector (New Englands Biolabs, Ipswich, MA, USA). For the construction of NT^Δ^CTmyc, the CT domain (codon 300–337) was deleted from NTCT in pBT by an *Eco* RI/*Bsp* E1 digest and subsequent insertion of the annealed HdCt primer pair as outlined above. Further cloning was as for the NTCTmyc construct.

The two N-terminal deletion constructs 315–358 and 315–369 were PCR amplified with an upper primer containing a *Xho* I site and lower primers with a *Hind* III restriction site, providing a Stop codon as well. Upper primer cTmyc up: 5′ GCG A***CT CGA G***CC CTG GAA ACA GTC C 3′; lower primers (cTmyc lo 358): 5′ GGT GGT ***AAG CTT*** CTA GTG GGG GCG C 3′ and (cTmyc lo 369): 5′ CC***A AGC TT***A AAT GGG CTG CTG ATC CTC G 3′. After the respective digest, the PCR products were inserted into pMAL-myc generated by an *Xho* I/*Hind* III digest of NTCTmyc in pMAL. The 295–317-myc construct was generated by insertion of an annealed primer pair with respective overlaps into *Xho* I/*Hind* III digested pMal-myc: CTmyc up 5′ ***TCG AG***A GCT TTT CGG ACG ACA ACA GCT CGA TAC AAT CCT CTC CTT GGC AGC GAG ACC AGC CCT GGA AAT **A** 3′ and CTmyc lo 3′ **C**TC GAA AAG CCT GCT GTT GTC GAG CTA TGT TAG GAG AGG AAC CGT CGC TCT GGT CGG GAC CTT TA***T TCG A*** 5′.

The Asf1-HA tagged region was derived from the pJG4-5 yeast vector providing an N-terminal HA-tag by PCR using the upper primer 5′ GGA GAT ***AGA TCT*** TAC CCT TAT GAT G 3′ with a ***Bgl* II** restriction site and the M13 reverse primer. The *Bgl* II/*Hind* III digested PCR product was inserted into the *Bam* HI/*Hind* III sites of the pMAL vector.

#### 2.5.2. Protein Expression

Maltose-binding protein MBP fusion proteins encoded by pMAL vectors and Glutathione S-transferase GST fusion proteins encoded by pGEX-2T vector [69] were expressed in *Escherichia coli* according to standard procedures [70,71,72]. To this end, pMAL constructs encoding the NTCTmyc variants and Asf-HA, respectively, and Su(H) (codons 288–594) in pGEX [11] were transformed into *E. coli* UT580 [71] and grown to log phase. Expression was induced with 1 mM IPTG at 18 °C overnight and bacteria lysed by French press. MBP fusion proteins were affinity purified using amylose beads (New Englands Biolabs, Ipswich, MA, USA) and eluted with maltose in 500 µL fractions as outlined in Ref. [73]. Likewise, the Su(H)-GST fusion protein was affinity purified on glutathione-sepharose 4B beads (GE Merck, Darmstadt, Germany) as described before [74]. The protein content of the fractions was measured with a Bradford assay and correct protein expression confirmed by PAGE.

#### 2.5.3. Myc-Trap Binding Assay

The Myc-trap binding assay was performed according to the manufacturer’s protocol (ChromoTek, Planegg-Martinsried, Germany) using anti-Myc-tag Nanobody/V_H_H coupled magnetic agarose resin (Myc-Trap^®^ Nanobodies). Purified proteins, Asf1-HA-MBP or Su(H)-GST were gently mixed with respective NTCTmyc-MBP peptides in a 1:1 ratio for about 10 min at 4 °C in a total volume of 500 µL of wash solution (150 mM NaCl, 50 mM Tris pH7.5, 0.1% SDS, 1 tablet of cOmplete^TM^ protease inhibitor per 10 mL (Roche Merck, Darmstadt, Germany)). Subsequently, 25 µL prewashed Myc-Trap magnetic beads were added per assay for 1 h at 4 °C under rotation. Beads were separated on a magnetic stand and washed three times before the elution of the proteins with 50 µL of 3× SDS-loading dye plus 0.125 M DTT. Probes were boiled for 10 min, spun briefly, before loading 15 µL of the supernatant onto SDS gel for electrophoresis and subsequent Western blotting. Blots were probed with the following antisera, mouse anti-Myc (1:500, 9B11 Cell Signaling Technology, Danvers MA, USA), rat anti-HA (1:500, 11867423001 Roche Merck, Darmstadt, Germany) and goat anti-GST (1:5000, 27,457,701 GE Healthcare, Munich, Germany), to detect the tagged H, Asf1 and Su(H) proteins, respectively.

### 2.6. In Vivo Protein Analysis

For in vivo protein detection on Western blots, proteins were extracted from 15 wandering third instar larvae each, homozygous for *H^gwt^*, *H*^Δ*CT*^ and *y^1^ w^67c23^*, exactly as outlined before [22]. PageRuler^TM^ Plus prestained protein ladder, 10–250 kDa, was used (Thermo Scientific, Waltham MA, USA), allowing blots to be cut at around 100 kDa. The upper part was treated with polyclonal rat anti-H antisera (h5, 1:250 [75]), the lower with monoclonal mouse anti-β-Tubulin A7 (1:500; developed by M. Klymkowsky, obtained from the Developmental Studies Hybridoma Bank DSHB, created by the NICHD of the NIH and maintained at The University of Iowa, Iowa City, IA, USA) as loading control. Secondary goat antibodies coupled to alkaline phosphatase (1:1000) were used for detection (Jackson Immuno-Research Laboratories via Dianova, Hamburg, Germany).

### 2.7. Isothermal Titration Calorimetry

Isothermal calorimetry was performed as described earlier [19,76]. *D. melanogaster* Asf1 protein (amino acids 1–154) was overexpressed and purified from bacteria using a modified pET-14p construct containing an N-terminal His-tag and C-terminal Strep-tag. Asf1 was purified using a combination of affinity and size exclusion chromatography. Hairless constructs (232–358, 315–358, 232–338 and 339–358) were overexpressed and purified as His-SUMO fusion proteins from a modified pET-28b+ vector and purified using affinity and size exclusion chromatography as previously described [20]. ITC experiments were carried out using a MicroCal VP-ITC calorimeter. All experiments were performed at 25 °C in a buffer composed of 20 mM HEPES pH 7.5 and 150 mM NaCl. The purified proteins were degassed and buffer-matched using dialysis and/or size exclusion chromatography. A typical experiment consisted of 100 μM Hairless in the syringe and 10 μM Asf1 in the cell. Protein concentrations were determined by UV absorbance at 280 nm. At least three independent experiments were performed. The data were analyzed using ORIGIN software and fit to a one-site binding model.

## 3. Results

### 3.1. Gene Engineering to Generate the H^ΔCT^ Allele, Specifically Lacking the CT Domain Only

Based on previous studies, the functions of the conserved domains of the Hairless protein were assigned to the binding of Su(H), the co-repressors Gro and CtBP and to nuclear translocation [7,11,12,13,14,22]. However, the function of the highly conserved CT domain, located directly C-terminal to the Su(H) binding domain NT, remained unknown (Figure 1a). To elucidate the role of the CT domain, we first generated an UAS-attB-*H*^Δ*CT*^ line and used it in tissue-specific overexpression studies via the Gal4-UAS system [45]. However, the results were inconclusive, since no conspicuous phenotypic differences were observed compared to wild-type H control (Appendix A), suggesting that there is little difference between the two transgenes. Nonetheless, ectopic expression leads to dramatically altered phenotypes and tissue loss, such that subtler functional differences may go unnoticed (Appendix A) [19,23,26,77,78,79,80]. Hence, we engineered a novel *H* allele *H*^Δ*CT*^, lacking the CT domain, for a detailed functional analysis of the CT domain on H activity. Since an intron is splitting the CT box in two parts, the cloning design was complex (Figure 1). In the first step, we removed the intron plus large parts of the CT box to eventually reintroduce the intron sequences including splice sites (Figure 1a,c). Hence, the resulting gene construct is identical to the wild type apart from the small deletion of 38 codons, covering most of the conserved CT domain (Figure 1b). After sequence verification, *H*^Δ*CT*^ genomic DNA was integrated into the *H^attP^* knockout line by genome engineering to generate the new *H*^Δ*CT*^ allele as outlined previously [43]. A stable line was generated after floxing the *white+* gene marker; it turned out that *H*^Δ*CT*^ is homozygous viable, indicating that the CT domain is not absolutely required for the repressive H function, whereas a loss of H activity results in larval/pupal lethality [17,43,81,82]. A Western Blot analysis revealed the expression of the shortened *H*^Δ*CT*^ protein isoform, including the smaller variant derived from internal ribosome entry, indicative of correct splicing (Appendix A) [75,80,83].

### 3.2. The H^ΔCT^ Allele Displays H Gain of Function Phenotypes

Homozygous *H*^Δ*CT*^ flies are viable and fertile. Only upon close inspection, the phenotypic alterations became apparent. As a control, we used *y^1^ w^67c23^* representing the genomic background of the stocks, as well as *H^gwt^*, which is a genome-engineered control harboring the genomic wild-type DNA [43]. Compared to these controls, we observed an increase in the number of microchaetae, the small mechano-sensory bristle organs that cover the fly thorax [59,84,85]. To analyze this phenomenon more precisely, scanning electron micrographs were taken and the microchaetae counted in a field between intrascutal suture and posterior dorsocentral macrochaetae (Figure 2a,b). The quantification showed a highly significant increase in microchaetae numbers in the *H*^Δ*CT*^ flies compared to the controls (Figure 2c). In contrast to macrochaetae, whose number and position is strictly defined, the number of microchaetae is also variable in wild-type flies, determined, e.g., by thorax size [59,84,85,86]. During pupal development, microchaetae are positioned in distinct rows; their spacing is controlled by lateral inhibition governed by Notch signal activity [59,87,88]. Accordingly, a downregulation of the Notch activity results in more densely spaced bristles, whereas its upregulation causes balding [12,82,89,90,91]. Apparently, the *H*^Δ*CT*^ allele displayed a mild gain of function phenotype in agreement with a slight downregulation of Notch activity.

The second remarkable phenotype concerns wing development, since a large fraction (81%) of *H*^Δ*CT*^ homozygotes developed additional wing vein material. Small veinlets appeared mostly within the marginal cell, i.e., between the first and the second longitudinal veins, and within the second posterior cell next to the posterior cross vein (Figure 3a,b). In about 3% of the flies, wings displayed several ectopic veinlets also within the submarginal cell, i.e., between the second and third longitudinal veins, and within the third posterior cell next to L5 (Figure 3a–c). The wing size appeared somewhat smaller than in the control; however, the differences were not significant. Similar to the spacing of microchaetae, vein formation is controlled by Notch signaling activity applying mostly to vein thickness but also to veinlet formation [92,93,94]. For example, heterozygous null alleles of *H* lack veins partially due to the gain of Notch activity, whereas Notch heterozygotes display thickened veins. A second wing phenotype results from a downregulation of Notch activity, namely notches in the wing margin, which the pathway derives its name from [1,82]. The cells giving rise to the wing margin form the dorso-ventral boundary of the wing anlagen and act as a source for morphogens regulating patterning and growth. These cells are fated by Notch activity, which induces the expression of several specific target genes, including Deadpan (Dpn) [93,95]. As no wing incisions were observed, we asked whether the *H*^Δ*CT*^ mutation does affect Dpn expression. To this end, we applied a mosaic analysis [61], generating homozygous *H*^Δ*CT*^ cell clones to be compared with homozygous wild-type cells for Dpn protein expression (Figure 3d–d“). As expected for a gain of H repressive activity, Dpn protein accumulation was lowered in *H*^Δ*CT*^ mutant cells compared to the sibling cells. In sum, the phenotypes displayed by the *H*^Δ*CT*^ homozygotes are in line with a stronger repressive activity, suggesting that the CT domain somehow attenuates normal H repressor function.

### 3.3. The H^ΔCT^ Allele Attenuates Cell Fating Defects

Apart from the above phenotypes concerning microchaetae and wing development, the *H*^Δ*CT*^ homozygotes appeared to be wild type. For example, the number of large bristles, i.e. the macrochaetae, was normal (Figure 4a–c). If the *H*^Δ*CT*^ allele were in fact a gain of H function, it should show a positive genetic interaction with a null allele of *H*. To test this idea, we performed crosses with *H^attp^* lacking any H activity [43] and compared the resultant phenotypes with *H^attp^* crossed to *H^gwt^* for a control. In the case of a gain of function, we expected an ameliorated phenotype in the *H^attp^*/*H*^Δ*CT*^ flies compared to the *H^attp^*/*H^gwt^* heterozygotes. *H* heterozygous flies display a dominant loss of bristle phenotype, affecting both micro- and macrochaetae. In addition to the loss of an entire mechano-sensory organ, sometimes, a transformation of the outer shaft to socket cell is observed, giving rise to a so-called double socket phenotype [43,58,91,96] (Figure 4d,d‘). In our analysis, we concentrated on the forty large bristles, as their number and position are a constant trait [84,85,86]. About fifteen macrochaetae were affected in *H^attp^*/*H^gwt^* control flies, on average, but only about nine in the *H^attp^*/*H*^Δ*CT*^ flies, in agreement with a significant gain of H activity (Figure 4e,f). A closer look, however, revealed that bristle loss was indistinguishable between *H^attp^*/*H*^Δ*CT*^ and control (Figure 4g). Accordingly, the phenotypic difference incorporated almost exclusively cell transformations, which were virtually absent in *H^attp^*/*H*^Δ*CT*^ heterozygotes (Figure 4e,h). This result was rather unexpected, as it may hint at a specific role for the CT domain during cell fate selection. Alternatively, dose differences could also be the explanation for these phenotypic differences. In this case, a minute increase in H activity may suffice to allow cell fate distinction but not sensory organ precursor selection.

### 3.4. Complex Genetic Interactions between Asf1 and Hairless

The above results demonstrate that the conserved CT domain somehow mitigates H activity, raising the possibility that it may serve as a binding site for respective factors. One binding partner of H to be considered in this context is Asf1. Earlier, it was shown that Asf1 promotes Notch repression together with H, albeit the molecular mechanisms are unknown [28]. The ectopic expression of *Asf1* within the eye anlagen was shown to reduce eye size, ameliorated by the concomitant loss of one *H* gene dose, suggesting that *Asf1* and *H* collaborate in repressing N activity [28]. In fact, *Drosophila* eyes are somewhat enlarged in *H^attP^* heterozygous null mutants and likewise upon tissue-specific downregulation of *H* activity during eye development by RNA interference (Appendix A). This phenotype is in accordance with an overactivation of the Notch pathway inducing growth [97,98,99]. On the other hand, induction of *H* in the eye anlagen confounds eye development, resulting in pupal lethality (Appendix A). In this instance, growth is inhibited, and cell death is induced [23,100,101,102]. Moreover, we observed a significant amelioration of the *Asf1*-induced small eye phenotype by a loss of H activity (Appendix A), in agreement with the published data [28]. Whether this apparent rescue is based on direct genetic interactions or on additive effects, however, remains unclear. Hence, we asked the question whether a downregulation of *Asf1* by RNA interference would allow for a more comprehensive analysis. We also included the other known co-repressors that bind H, namely Gro and CtBP, in the hope of gaining additional functional insights (Appendix A). However, the results were inconclusive because the RNAi-mediated downregulation of *Asf1* in the developing eye resulted in smaller eyes similar to its ectopic expression. Perhaps the loss of Asf1 activity induced cell death. This explanation seems likely, firstly because *Asf1* homozygosity was reported to be cell lethal [28], and secondly because the respective eyes display typical signs of cell death (Appendix A). Specificity of the Asf1-RNAi was confirmed by the concurrent induction of Asf1, and these flies displayed near wild-type eyes (Appendix A). Quantification of the phenotypes indicated some rescue of the Asf1-RNAi-induced small eye phenotype by loss of H activity, which again could be an additive effect or a specific counteraction of a gain of Notch activity against cell death induction. Moreover, downregulation of either co-repressor Gro or CtBP slightly enhanced the eye phenotype as well, perhaps due to increased cell death or a further block of tissue growth (Appendix A).

### 3.5. Mapping the H-Asf1 Interaction Domain by Yeast Two-Hybrid Assays

Earlier reports demonstrated a physical interaction of the full-length Asf1 and H proteins by pull-down assays [28]. We aimed to define the Asf1 interaction domain in H by yeast two-hybrid analyses using a number of deletion constructs for H. The analysis, however, was hampered by the extremely weak interactions between the two when compared to the positive control Su(H). We were unable to see any binding between the full-length proteins and only extremely weak binding of Asf1 to the isolated NTCT domain even after prolonged exposure (Figure 5). The NTCT domain is highly conserved among insect species and is even found in the H protein from the arthropod *D. pulex* [23,24,103]. NTCT can be split into N-terminal NT and C-terminal CT domains; the NT domain is sufficient for the binding to Su(H), whereas no previous function has been assigned to the CT domain [19,20,23] (Figure 5). We noted a weak binding of Asf1 to the isolated CT domain and a rather robust binding to the NTCT^Δ^NT peptide. Specificity of the interaction was confirmed by a lack of binding to the NTCT^Δ^CT peptide, which, on the contrary, bound well to Su(H) as expected (Figure 5). In fact, Asf1 is the first protein identified to potentially contact this conserved region of the H protein. The weak interaction between the two, however, is puzzling in light of the previous reports [28]. One reason could be the high conservation of Asf1 proteins in eukaryotes. Even the Asf1 orthologs from yeast and fly share almost 60% sequence identity in the active domain (residues 1–156) (Appendix A) [104,105]. Potentially, the endogenous yeast Asf1 outcompetes the *Drosophila* Asf1-AD protein expressed from pJG, thereby lowering productive reporter activation. Hence, we sought other technical approaches for mapping the H–Asf1 interface.

### 3.6. Interaction Assays Using Tagged Hairless and Asf1 Proteins

Next, we generated several Hairless NTCT variants that were tagged with Myc, as well as full-length *Drosophila* Asf1 tagged with HA. The respective proteins were expressed in bacteria as MBP fusion proteins and purified by affinity chromatography. As a positive control, a GST Su(H) fusion protein containing the H-binding C-terminal domain CTD was used [11]. For binding assays, the purified Asf1 or Su(H) proteins were mixed with H peptides and the Myc-tagged H proteins trapped with magnetic beads. After washing and elution, the eluates were analyzed in Western blots. The presence of the respective proteins in the H precipitates was probed with antibodies directed against myc, HA or GST (Figure 6).

As predicted from the above experiments, we found robust binding of Asf1 to NTCTmyc but not to NT^Δ^CTmyc, whereas both peptides co-precipitated with Su(H) as expected (Figure 6a,b). To map the binding domain more exactly, we subdivided the conserved CT domain into several shorter peptides, either containing the most highly conserved sequences only (aa 295–317) or starting further in the C-terminal, including conserved basic residues (aa 315–358 and aa 315–369) (Figure 6c,d; compare with Figure 1b). Unexpectedly, only the two C-terminally extending peptides bound to Asf1, despite their moderate expression, whereas the small peptide containing the N-terminal part of the conserved CT domain failed to do so (Figure 6c). From the current analysis, we tentatively map the H residues involved in Asf1 binding to amino acid residues 315–338. This is based on the assumption that (1) fragmentation does not interfere with higher order protein structure (i.e., every peptide assumes its correct fold) (Appendix A) and (2) that the binding site is restricted to a few isolated amino acids.

### 3.7. Thermodynamic Analysis of the Hairless and Asf1 Interaction

To provide an orthogonal approach to validate H/Asf1 binding, as well as further map the interacting regions and quantitatively measure binding, we used isothermal titration calorimetry with purified recombinant proteins of H and Asf1. As shown in Figure 7 and Table 1, H (232–358) bound to Asf1 (1–154) with ~4 µM affinity and a 1:1 stoichiometry. Similarly, H (315–358) bound Asf1 (1–154) with a comparable K_d_ of ~2 µM. However, when we further truncated our H construct to residues 232–338 or 339–358, respectively, we were no longer able to detect binding by ITC. Taken together, these data confirm the above results: the domain containing aa 315–338 appears absolutely required for the binding of Asf1; however, it is not sufficient on its own to support binding. Only H (315–358) was firmly established to contain the Asf1 binding domain.

### 3.8. The Asf1-Induced Small Eye Phenotype Is Enhanced in H^ΔCT^ Background

Earlier work suggested that Asf1 overexpression promoted Notch target gene repression most likely via Hairless [28]. Accordingly, an increased repression of Notch activity during eye development resulting from ectopic Asf1 protein could affect eye growth, resulting in smaller eyes. If indeed this phenotype depended on the combined activity of Asf1 and H, it should no longer be observed in the *H*^Δ*CT*^ background, where Asf1–H complexes cannot form due to the lack of the Asf1 binding site in H. To address this idea, we combined *H*^Δ*CT*^ with the ey-Gal4 UAS-Asf1 expression line and noted enhanced phenotypes instead of a rescue, independent of sex (Figure 8). Quantification of the eye size confirmed the visual impression; whereas the eye size of *H*^Δ*CT*^ matched the control, the ectopic expression of Asf1 during eye development reduced eye size by roughly ten percent, and more than twenty percent in the *H*^Δ*CT*^ background (Figure 8), again independent of sex. These data clearly illustrate genetic interaction between H and Asf1—yet, in an unexpected manner. Evidently, H ameliorates Asf1 repressive activity, since in the absence of H binding, Asf1 repressive activity increased, resulting in reduced growth (Figure 8).

## 4. Discussion

In this work, we applied genome engineering to generate a novel *H*^Δ*CT*^ allele, encoding an H protein lacking the conserved CT domain, allowing us to specifically address CT’s biological function. Unexpectedly, *H*^Δ*CT*^ flies were viable without restriction, suggesting that the CT domain is not essential for H repressor function during Notch signaling. Moreover, the subtle gain of function phenotypes displayed by *H*^Δ*CT*^ flies imply that the CT domain somehow attenuates normal H repressor activity. Of note, these gain of function phenotypes are largely restricted to the process of lateral inhibition during the selection of mechano-sensory organ precursors and the specification of wing veins (Figure 2 and Figure 3). During lateral inhibition, cells are selected from a field of equipotential cells with the same fate, i.e., pro-neural or pro-vein fate, to differentiate correspondingly. The remaining cells refrain from the respective fate due to Notch signaling activity, involving a transcriptional feedback loop that amplifies small differences between the cells [87,106,107,108]. In this context, H protects the presumptive sensory organ precursors or pro-vein cells from spurious Notch signals [109,110,111]. Our data suggest that *H*^Δ*CT*^ is more potent in doing so, thereby allowing more cells to choose the primary fate, resulting in extra sensory organs, as well as ectopic veinlets (Figure 3a–c). This may reflect a specific activity of *H*^Δ*CT*^, for example, in feedback regulation, or just a slightly stronger activity overall. A simple explanation could be that CT deletion helps stabilize the H protein. Dose sensitivities for Notch signaling components including H are well described [17,82,112,113]. Notably, bristle numbers are highly susceptible to phenotypic plasticity due to genomic fluctuations, where H plays an important role [114,115,116]. Our recent studies, however, give no indications that the CT domain might contribute to H stability or might contain a degron (Appendix A) [32,62]. An additional hint of a specific regulatory role of the CT domain comes from the observation that the *H*^Δ*CT*^ heterozygotes did not display the characteristic double socket phenotype of the *H* heterozygotes (Figure 4). This phenotype is caused by a transformation of prospective shaft into socket cells [91,96]. Shaft and socket cells are siblings derived from an asymmetric cell division; they are differentially fated via a Notch signal in the presumptive socket cell, which induces a specifying, auto-regulatory Su(H) activity [96,117,118,119]. In the *H* heterozygotes, the Su(H) auto-activation also takes place in presumptive shaft cells, causing a fate change, and hence, apparent transformation into a socket cell [89,96,118,119]. The *H*^Δ*CT*^ heterozygous condition, however, allows for shaft cell development, i.e., the Su(H) autocatalytic feedback loop is restricted to the presumptive socket cell. Hence, *H*^Δ*CT*^ clearly differs from the wild type regarding its regulatory capacity. Whether this is based on dose or activity differences remains to be established.

Our in vitro protein interaction experiments demonstrated that *H*^Δ*CT*^ lacks Asf1 binding (Figure 6a,d). The ^Δ^CT deletion comprises only 38 amino acids, of which codons 315–338 are indispensable for H–Asf1 binding. The minimal sequences sufficient for Asf1 binding are contained within 44 amino acids extending further C-terminal up to codon 358, with an overlap of 23 highly conserved amino acids (Figure 1b) [23]. Splitting this region apart did not support any Asf1 binding, suggesting an underlying structural entity; however, structural analyses of H–Asf1 complexes are currently lacking. In silico models of the H NTCT domain predict a β sheet followed by an α helix, none of which are sufficient for the binding (Figure 7 and Appendix A) [120]. As the CT domain does not influence the binding of Su(H), the repressor complex formation itself should be unaffected. This raises the possibility that CT may influence the binding dynamics of the repressor complex on chromatin, for example, via Asf1. Earlier reports showed that Asf1 contributes to the repression of Notch target genes, since RNAi-mediated knock-down caused an upregulation of Notch target genes, notably of *E(spl)m3, E(spl)m7* and *E(spl)mγ* [28]. Moreover, local induction of Asf1 in the eye and wing anlagen was accompanied by tissue loss, in line with a downregulation of Notch growth-promoting activities [28,97]. The physical interactions between H and Asf1 prompted a model, whereby Asf1 was recruited to the Notch target genes via H, promoting transcriptional silencing [28]. Clearly, based on our studies herein, the regulation mediated by H and Asf1 is more complex. Overall, our genetic data reveal a mutual negative relationship between H and Asf1, i.e., both attenuate each other’s repressive activity in specific developmental contexts. On the one hand, growth retardation induced by the ectopic expression of Asf1 was enhanced in the *H*^Δ*CT*^ background, implying that, normally, H impairs Asf1 repressor activity in the context of overexpression (Figure 8). Perhaps H, in contrast to *H*^Δ*CT*^, is able to sequester ectopic Asf1, thereby reducing the detrimental effects on growth. On the other hand, Asf1-binding-defective *H*^Δ*CT*^ is a stronger repressor than wild-type H, suggesting that Asf1 binding may reduce H repressor activities (Figure 2 and Figure 3). Conceivably, Asf1 impedes the dynamics of H–Su(H) repressor complex exchange by chromatin modifications [28,34,38]. The counteractive function of Asf1 may be restricted to the transient period of interactions between the H–Su(H) repressor complex and the promoter region before stable silencing occurs. We envisage that during the process of lateral inhibition, when cell fates are not yet fixed, a flexible regulatory input is needed to allow the transcriptional feedback loop to take place for signal amplification, eventually generating the differential outcome [87,106]. The activity of Asf1, however, may induce premature stable chromatin states no longer amenable to a dynamic regulation. Alternatively, the recruitment of Asf1 to the Su(H) repressor complex is primarily mediated by Skip (Bx42 in *Drosophila*) and, to a lesser degree, by H [28]. In mammalian cells, Skip binds to the Su(H) homolog CBF1, acting as a hub for activating and repressing co-factors [121,122]. Subsequently, Asf1 may spread into the transcribed gene region, increasing nucleosome density, in accordance with its well-documented role in nucleosome assembly in ORFs during gene silencing [42]. Accordingly, RNAi-mediated depletion of H caused Asf1 disappearance not only from the promoter regions but also from the ORF of *E(spl)m7* and *E(spl)m3* genes [28]. Overall, our work adds to the growing understanding of the concerted action of transcriptional regulators, histone modifiers and chromatin remodelers to organize chromatin and transcription.

## Figures and Tables

**Figure 1 genes-14-00205-f001:**
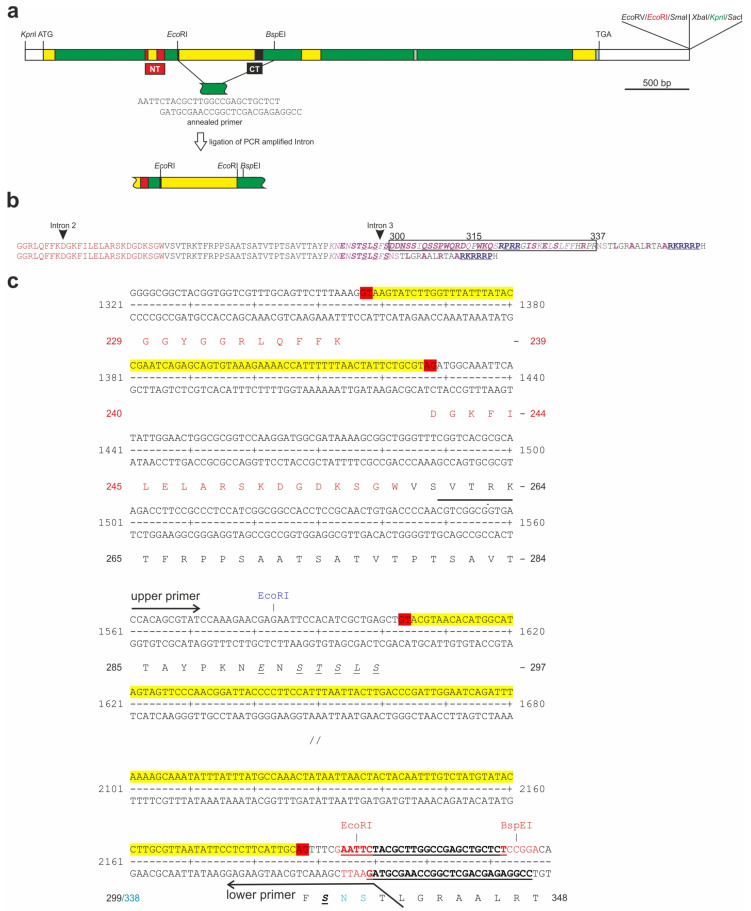
Generation of *H*^Δ*CT*^ flies by genome engineering. (**a**) Full-length genomic *H* DNA was cloned into pBT between *Kpn* I and *Eco* RV polylinker sites [17]. Coding sequences are shown in green, intron sequences in yellow and leader and trailer sequences in white. Sequence coding for Su(H) interaction domain is shown in red (NT, N-terminal domain); the conserved C-terminal domain (CT) is depicted in black. The *Eco* RI site (red) was subsequently removed from the polylinker by re-ligation after *Eco* RV and *Sma* I digestion. A *Kpn* I site (green) was added at the 3′ end by *Xba* I and *Sac* I digest followed by ligation of an annealed primer with compatible sticky ends. This was necessary for final cloning in the *Kpn* I site of the pGE-attB-GMR vector [44]. The CT domain, including the third intron, was deleted by an *Eco* RI/*Bsp* EI double digest. A primer pair with *Eco* RI/*Bsp* EI sticky ends as well as the deleted sequences of the third intron was annealed and ligated into the opened construct; see (**c**). The third intron was PCR amplified with an upper primer 5′ of the endogenous *Eco* RI site and a 3′ lower primer containing an *Eco* RI site and splice sites; see (**c**). The construct was then *Eco* RI digested, and the *Eco* RI digested PCR amplificate containing the third intron was reintroduced. Afterward, the genomic clone was ligated into the pGE-attB-GMR vector as *Kpn* I fragment and inserted into the *H^attP^* founder line to generate *H*^Δ*CT*^ flies [43,44]. (**b**) Relevant parts of the H amino acid sequence containing the NTCT domain are shown. The upper amino acid sequence displays the wild-type H with the Su(H) binding motive NT in red and the conserved CT domain in italics; magenta residues are conserved in *A. mellifera*, the bold ones are identical, and underlined ones also in *D. pulex* [23,24]. Conserved basic sequences are shown in purple [22]; those identical in the honeybee are underlined. Framed residues (300–337) are deleted in *H*^Δ*CT*^. (**c**) Relevant genomic sequence in *H*^Δ*CT*^ DNA with restriction sites used for cloning. Introns are marked in yellow. The introns conform to the GT-AG rule (marked in red). Underneath the amino acid sequence: in red, the Su(H) interaction domain NT; in italics and underlined, the remaining conserved residues between *Drosophila* and *Daphnia*. The newly introduced *Eco* RI restriction site (red) did not change the asparagine and serine codons (N,S in light blue) from wild type. Arrows indicate the position of the primer pair for intron amplification. The 5′ start bases of the lower primer correspond to wild-type sequences deleted after *Eco* RI digestion of the PCR amplification product. The sequences between the *Bsp* EI and *Eco* RI restriction sites were introduced with the annealed pair of primers (bold and underlined) indicated in (**a**).

**Figure 2 genes-14-00205-f002:**
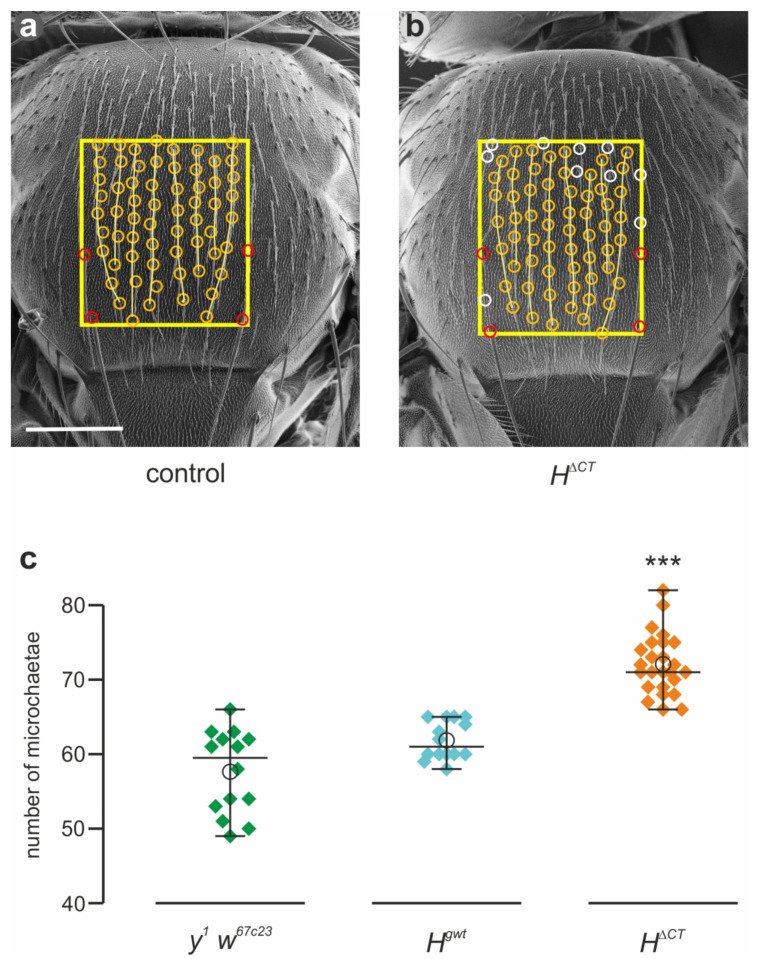
Gain of function bristle phenotypes in *H*^Δ*CT*^ homozygotes. (**a**,**b**) Scanning electron micrographs of typical thoraces from *y^1^ w^67c23^* (control) and from *H*^Δ*CT*^ female flies, respectively. Microchaetae (encircled in yellow) were counted in the outlined area between intrascutal suture and posterior dorsocentral macrochaetae (encircled in red). Microchaetae form eight regular rows (yellow lines) in the control *y^1^ w^67c23^* flies (**a**), whereas they occur at irregular positions (encircled in white) and are more densely spaced in *H*^Δ*CT*^ (**b**). Scale bar, 200 µm in (**a**,**b**). (**c**) Microchaetae counts from *y^1^ w^67c23^* and *H^gwt^* for control (*n* = 14) and from *H*^Δ*CT*^ (*n* = 25) were plotted. Center lines show the medians, empty circles the means; whiskers extend 1.5 times the interquartile range from the 25th and 75th percentiles, respectively. Significance was determined with ANOVA two-tailed Dunnett’s test for multiple comparisons; significant differences are observed between both the controls and *H*^Δ*CT*^ (*** *p* ≤ 0.001).

**Figure 3 genes-14-00205-f003:**
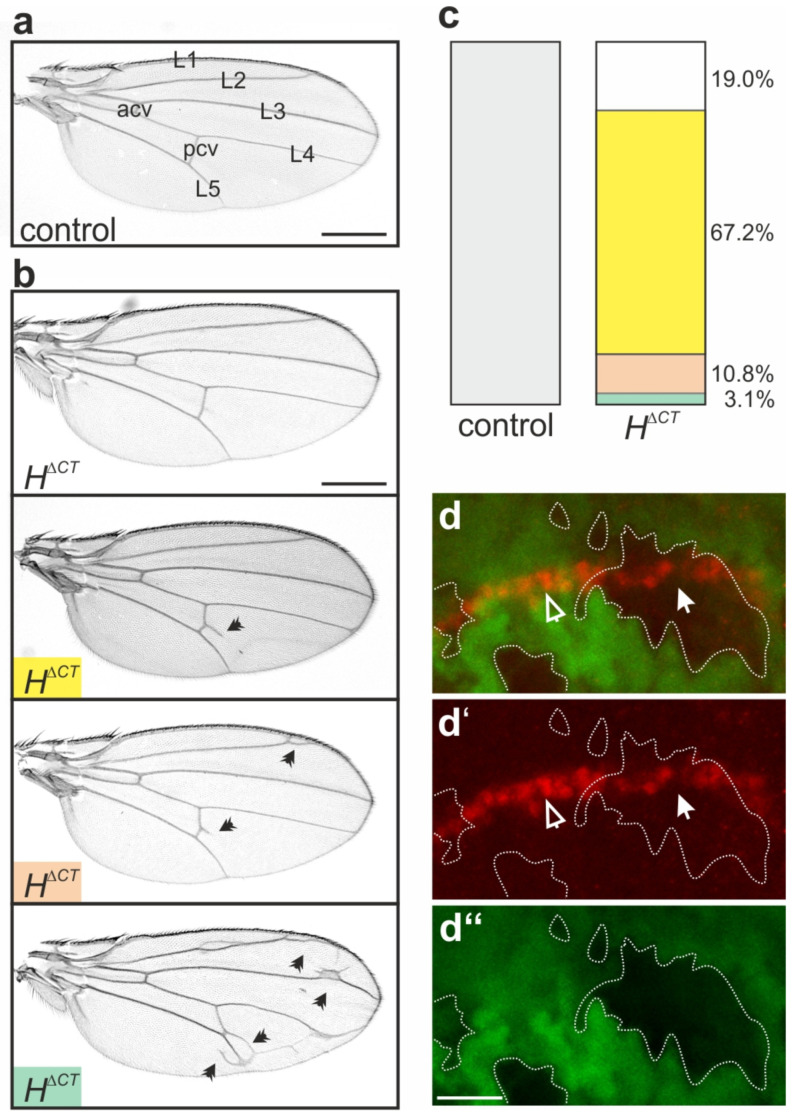
Wing defects in homozygous *H*^Δ*CT*^ flies. (**a**) Wing from control *y^1^ w^67c23^* female fly. Note the typical five longitudinal veins (L1–L5) and the anterior and posterior cross-veins (acv, pcv), respectively. (**b**) Typical examples of wings from homozygous *H*^Δ*CT*^ female flies; the majority display one or several ectopic veinlets (arrowheads point to examples). The wing size is somewhat smaller than the control; however, the difference is not significant. Scale bar, 500 µm in (**a**,**b**). (**c**) Graph summarizing the range of wing phenotypes as displayed and color coded in (**b**), control *n* = 30; *H*^Δ*CT*^ = 195. Total number of wings analyzed is given. (**d**–**d“**) Mosaic analysis; wild-type and heterozygous cells are marked by GFP (green, **d**,**d“**). The *H*^Δ*CT*^ mutant cell clones are unmarked (outlined by a dashed line). Dpn protein expression along the dorso-ventral boundary is shown in red (arrowheads in **d**,**d‘**). Compared to wild-type clones (open arrowhead), Dpn protein is lowered in the *H*^Δ*CT*^ mutant cell clones (closed arrowhead). Scale bar, 20 µm.

**Figure 4 genes-14-00205-f004:**
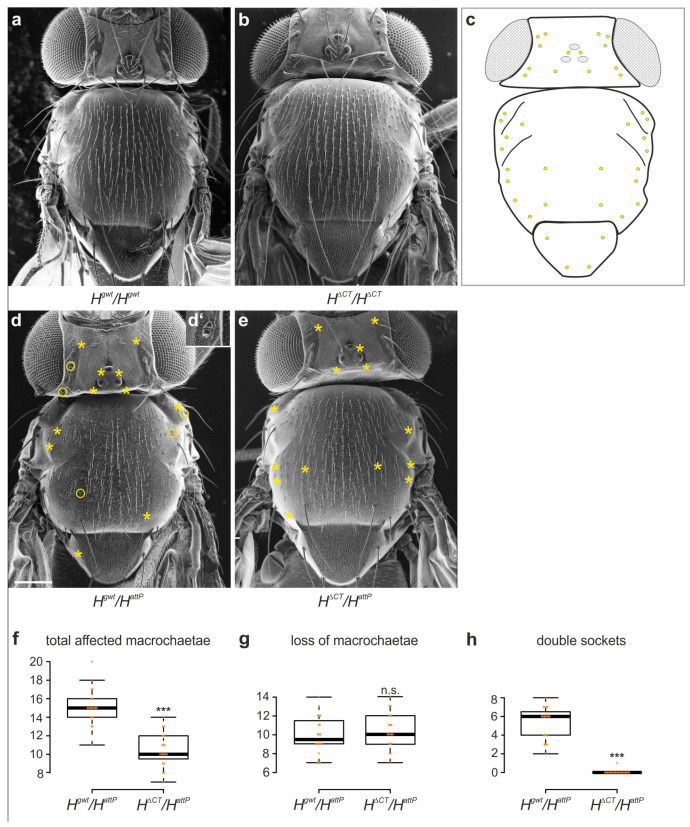
Macrochaetae formation in hemizygotes. (**a**,**b**) Scanning electron micrographs from homozygous controls *H^gwt^*/*H^gwt^* (**a**) and *H*^Δ*CT*^/*H*^Δ*CT*^ (**b**), respectively. No defects in macrochaetae observed (*n* = 14,20). (**c**) Schematic drawing showing the 40 defined positions of macrochaetae in the wild type [86]. (**d**,**e**) Typical examples of hemizygous female flies *H^gwt^*/*H^attP^* (**d**,**d‘**) and *H*^Δ*CT*^/*H^attP^* (**e**). Asterisks mark the absence, circles a shaft to socket transformation (example of a resultant double socket is shown enlarged in (**d‘**)). Scale bar, 200 µm in (**a**,**b**,**d**,**e**). (**f**–**h**) Quantification of macrochaetae misdevelopment (*n* = 20) as indicated. Box blot limits indicate 25th and 75th percentiles; whiskers extend 1.5 times the interquartile range; center lines show the medians (assembled by BoxPlotR). All values are indicated with orange dots, including outliers. Significance was determined with Student’s *t*-test (*** *p ≤* 0.001).

**Figure 5 genes-14-00205-f005:**
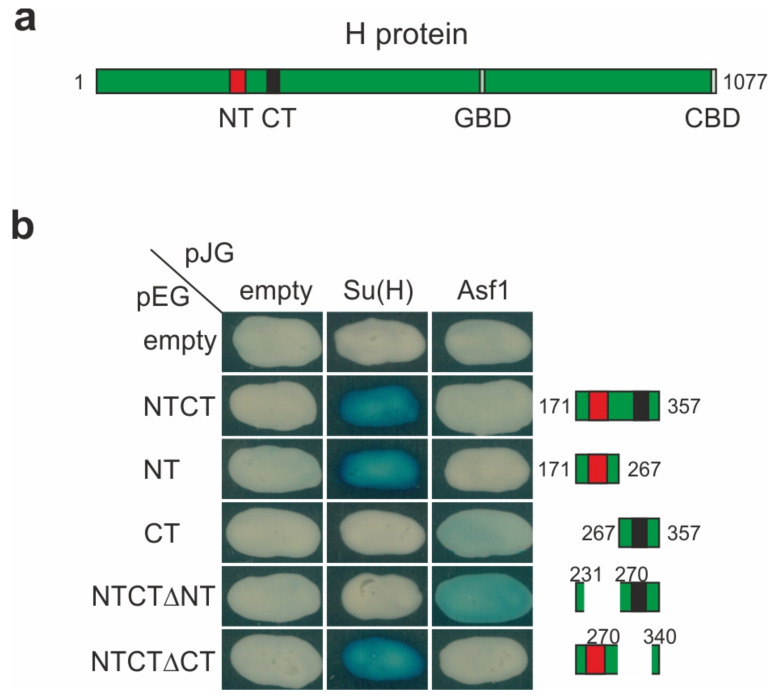
Asf1–H interaction revealed by yeast two-hybrid interaction assays. (**a**) Illustration of H protein structure. NT domain (red) includes the Su(H) binding domain. CT domain (black) is highly conserved and binds to Asf1 (shown in (**b**)). GBD, Groucho-binding domain; CBD, binding domain for CtBP. Numbering refers to amino acids of full-length H protein [17,75]. (**b**) Interaction tested between Asf1 and specific fragments of Hairless as indicated in the schemes. Size of fragments is given in codons for NTCT, NT and CT; size of deletion within NTCT given in codons for NTCTD^Δ^NT and NTCT^Δ^CT. Empty vectors (pEG, pJG) served as negative and Su(H) as positive controls. Protein interaction is revealed by blue coloration of yeast colonies. Note the different incubation times to observe conspicuous coloration: overnight for the Su(H)-H interactions, however, 48 h incubation for the Asf1–H interactions.

**Figure 6 genes-14-00205-f006:**
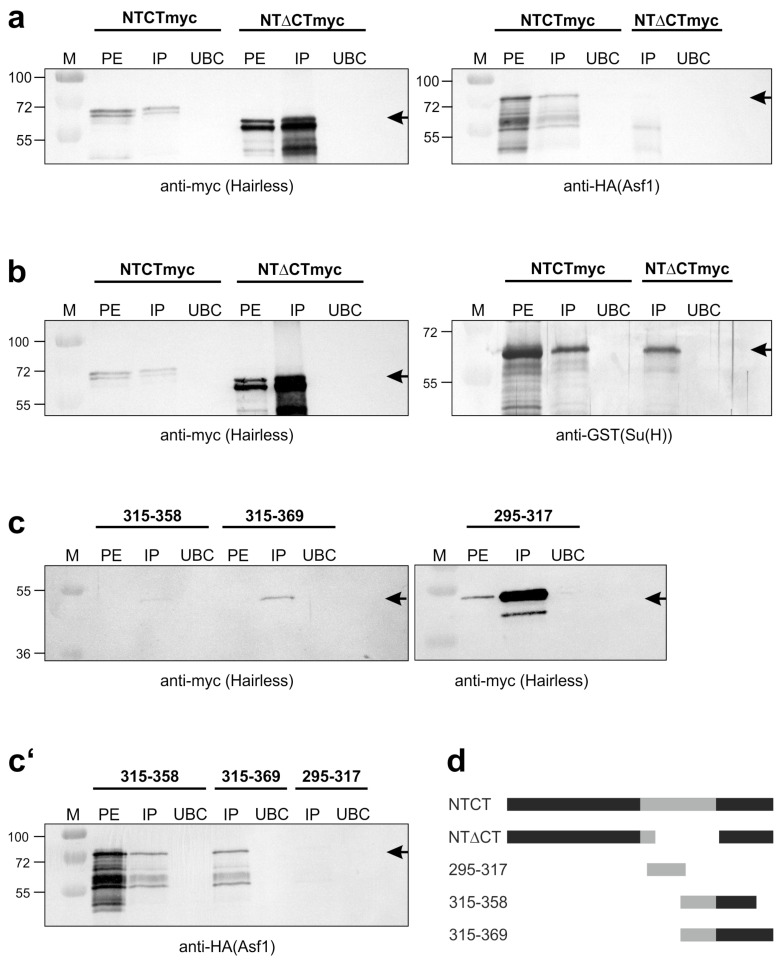
Mapping the Asf1–H interaction by co-immunoprecipitation experiments. (**a**–**c‘**) H peptides as indicated were Myc-tagged to allow for Myc-trap co-immunoprecipitation. Full-length HA-tagged Asf1 was used as binding partner; GST-tagged Su(H) served as control. Precipitates were detected with antibodies directed against Myc for H, against HA for Asf1 and against GST for Su(H) as indicated. (**a**) Asf1 is present in complexes with NTCT but not with NT^Δ^CT (corresponding to NTCT ^Δ^CT; see (**d**)), despite its strong expression. (**b**) As expected, Su(H) is found in complexes with NTCT as well as with NT^Δ^CT. (**c**,**c‘**) Asf1 is detected in precipitates together with the two C-terminal fragments 315–358 and 315–369 but not with the 295–317 peptide containing conserved CT sequences. (**d**) Schematic of the constructs used for the interaction mapping. The region of highest conservation designed CT domain is shown in light gray. Numbers correspond to codons contained within the respective constructs; NT^Δ^CT lacks codons 300–337. M: protein standard, PE: protein extract, IP: precipitation, UBC: unbound control.

**Figure 7 genes-14-00205-f007:**
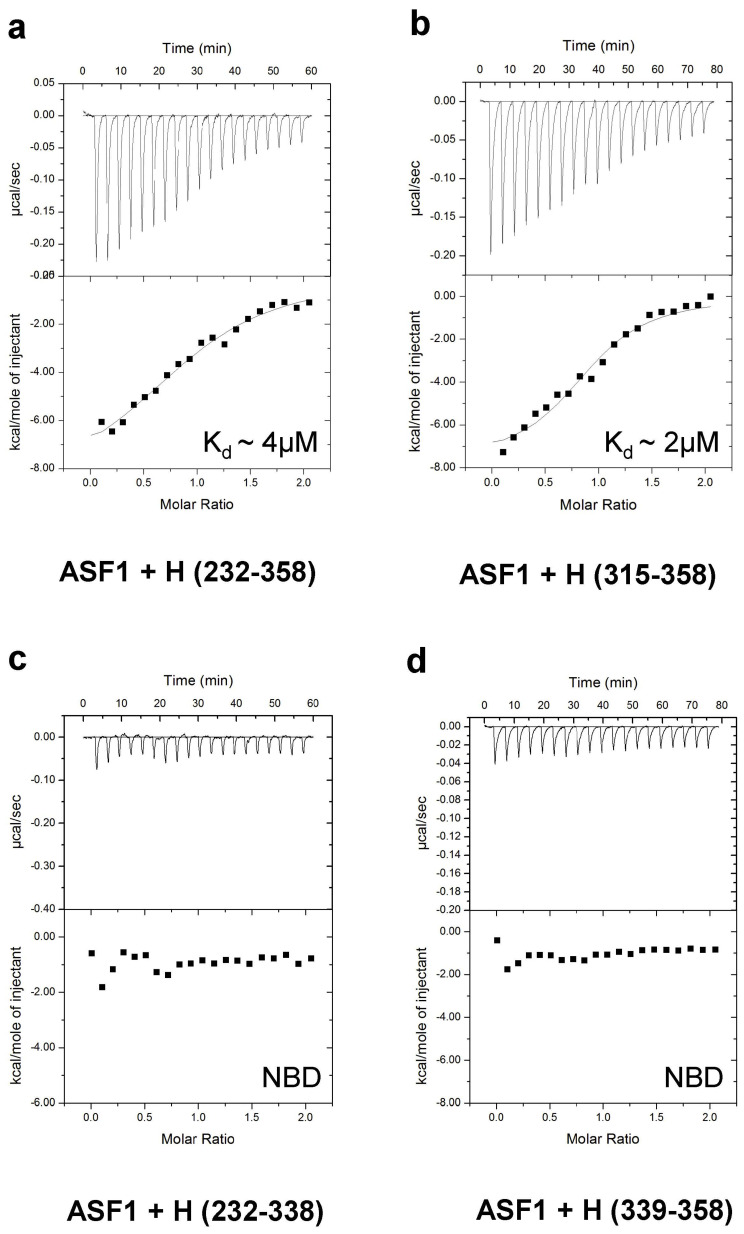
Mapping the Asf1–H interaction by isothermal titration calorimetry ITC. ITC binding analysis of H–Asf1 complexes. Representative thermograms showing the raw heat signal and non-linear least-squares fit to the integrated data for binding reactions of *Drosophila* Hairless and Asf1. Each ITC experiment was performed at 25 °C with 20 injections, and the average dissociation constants (K_D_) are shown. (**a**) Asf1 (1–154) binds to H (232–358) with ~4 µM affinity. (**b**) Asf1 (1–154) binds to H (315–358) with ~2 µM affinity. (**c**,**d**) No binding was detected between Asf1 (1–154) and H (232–338) and H (339–358), respectively. NBD = No Binding Detected.

**Figure 8 genes-14-00205-f008:**
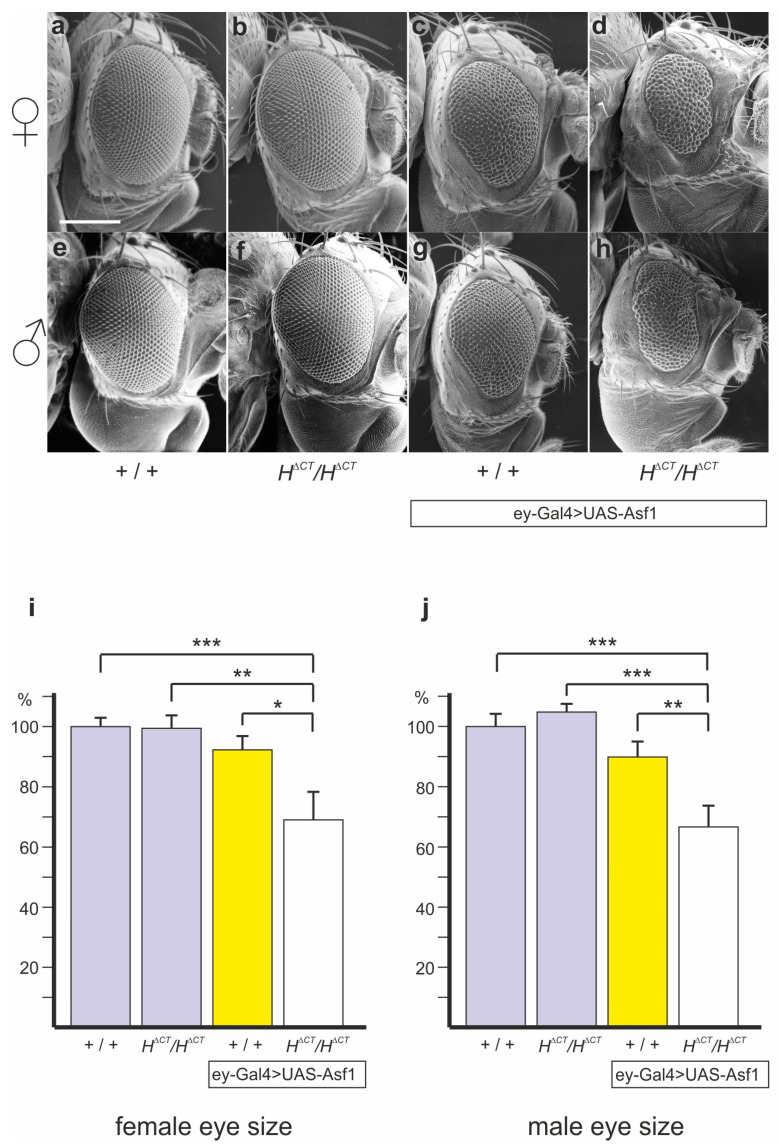
Eye size reduction induced by ectopically expressed Asf1 is enhanced in the *H*^Δ*CT*^ background. (**a**–**h**) Scanning electron micrographs from female and male fly heads of the indicated genotype. Note the eye size reduction upon tissue-specific induction of *Asf1* (**c**,**d**,**g**,**h**). Scale bar, 200 µm (**a**–**h**). (**i**,**j**) Quantification of the eye size from five female flies of the indicated genotype using Image J. Controls are the homozygous *y^1^ w^67c23^* (**a**) and *H*^Δ*CT*^ flies (**b**) (lilac bars). Eye-specific expression of *Asf1* reduced eye size (**c**) (yellow bar), which was significantly enhanced in the *H*^Δ*CT*^ background (**d**) (white bar). Significance was determined with ANOVA two-tailed Tukey–Kramer test for multiple comparisons (*** *p* ≤ 0.001; ** *p* ≤ 0.01; * *p* ≤ 0.05). Genotypes: *y^1^ w^67c23^*/*y^1^ w^67c23^* (**a**), *y^1^ w^67c23^*/*y^1^ w^67c23^*; *H*^Δ*CT*^/*H*^Δ*CT*^ (**b**), *y^1^ w^67c23^*/+ or Y; ey-Gal4 UAS Asf1/+ (**c**,**g**), *y^1^ w^67c23^*/*y^1^ w^67c23^*; ey-Gal4 UAS Asf1/CyO; *H*^Δ*CT*^/*H*^Δ*CT*^ (**d**), *y^1^ w^67c23^*/Y (**e**), *y^1^ w^67c23^*/Y; *H*^Δ*CT*^/*H*^Δ*CT*^ (**f**), *y^1^ w^67c23^*/Y; ey-Gal4 UAS Asf1/CyO; *H*^Δ*CT*^/*H*^Δ*CT*^ (**h**).

**Table 1 genes-14-00205-t001:** ITC binding for complexes formed between H and ASF1.

H	ASF1	*K* (M^−1^)	*K_d_* (µM)	Δ*G°* (kcal/mol)	Δ*H°* (kcal/mol)	*−T*Δ*S°* (kcal/mol)
232–358	1–154	2.5 ± 0.6 × 10^5^	4.2	−7.4 ± 0.1	−8.6 ± 0.3	1.2 ± 0.4
315–358	1–154	6.8 ± 2.6 × 10^5^	1.7	−7.9 ± 0.3	−8.7 ± 0.2	0.8 ± 2.2
232–338	1–154	NBD	---	---	---	---
339–358	1–154	NBD	---	---	---	---

All experiments were performed at 25 °C. NBD represents no binding detected. Values are the mean of at least three independent experiments, and errors represent the standard deviation of multiple experiments.

## Data Availability

Not applicable. All data are contained within the manuscript.

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
