# Peer review of "Genetic and Molecular Interactions between HΔCT, a Novel Allele of the Notch Antagonist Hairless, and the Histone Chaperone Asf1 in Drosophila melanogaster"

_genes, 2023, doi:10.3390/genes14010205_

Round 1

Reviewer 1 Report

Genetic and molecular interactions between HDCT a novel allele 2 of the Notch antagonist Hairless, and the histone chaperone 3 Asf1 in Drosophila melanogaster

In their manuscript Maier et al. present their analysis of genetic and molecular interactions of a novel allele of the Notch antagonist Hairless, HDCT, and the histone chaperone Asf1.

Initially the authors generated the HDCT allele by genome engineering. This allele is specifically lacking the CT domain of Hairless. Since the activity of Hairless can be monitored by the analysis of the number of microchaetae and alteration of the wing vein patterns, they carefully analysed the activity in these different tissues. An increase of the microchaetae number and additional veinlets argue for a gain of function phenotype of the HDCT allele. In homozygous HDCT cell clones Dpn expression was lowered at the dorso-ventral boundary of wing discs, again arguing for a stronger repressive activity. In the next set of experiments the genetic interaction of Hairless with Asf1 was analysed using different approaches. Via overexpression of Asf1 in the eye combined with alterations in Hairless activity it was shown that Asf1 and Hairless collaborate in repressing Notch activity. Mapping of the Asf1-Hairless interaction domain using yeast two-hybrid analysis turned out to be difficult due to the extremely weak interactions. Using tagged Hairless and Asf1 proteins it was possible to map amino acid residues 315-338 of Hairless involved in Asf1 binding. Also thermodynamic analysis using isothermal titration calorimetry proved that amino acid residues 315-338 are absolutely required for the binding of Asf1 to Hairless. Finaly the authors showed that the Asf1 induced small eye phenotype is enhanced in a HDCTbackground demonstrating genetic interaction between Hairless and Asf1, albeit not in the expected manner.

This study provides new insides concerning the genetic and molecular interactions between the novel Hairless allele HDCT and the histone chaperone Asf1. Since not so much was known up to now concerning the function of Asp1 in Notch signaling, the data presented here are a big step forward in this understanding and build the groundwork for further investigations.

The data of the paper are very good presented and the experiments justify all the conclusions drawn.

Minor comments:

line 204: the highlighted sequence in the primer is a XhoI restriction site, not a SalI restriction site.

The Figure 1a might be enlarged a bit (maybe as broad as 1b) to recognize the sequences better.

It might be good to get an idea about the size of the Hairless genomic fragment (maybe also in the figure legend, not only in Materials and Methods).

Author Response

Thank you for carefully reading the manuscript and for the helpful comments.

We enlarged Figure 1a, and included a scale bar. Moreover, we changed the name of the restriction enzyme (SalI -> XhoI) in the Materials and Methods section.

Reviewer 2 Report

This work focuses on a genetic analysis of the removal of a significant portion of a conserved portion of the Hairless protein called CT.  Hairless is an important factor in the conserved Notch pathway.  The paper analyses the HDCT mutant in Figures 2-4 and then the interaction between H and Asf-1 in Figures 5-7.  In Figure 8 they analyze a genetic interaction between HDCT and Asf-1.  

The paper is a bit difficult to read because of what seems to be contradicting data and expectations.  I think the paper should be publish when some of the following considerations have been addressed.  

First the authors refer to the GAL4UAS system as resulting in over expression.  They need to provide evidence of this.  When expression of proteins are assessed quantitatively with the GAL4UAS system it is found that they are not generally over expressed, but are definitely expressed ectopically.  If they have not demonstrated this they can not claim over expression even though it is a common unproven assumption made in the field.

Second the content of Figures 2 and 4 should be more united.  The data in Figure 4 does not have controls that I think are located in Figure 2.  Figure 4 should have a wild-type, Hgwt/Hgwt, HDCT/HDCT controls at a minimum.  It is very difficult to get what is going on without the controls.  

Third the authors should consult the alpha fold project for predicted structures in the H CT region.  Also I am not very convinced about the binding studies and whether it shows that Asf-1 binds to anything in the CT domain.  In the HDCT mutant both the CT and more C-terminal aa are removed, and in the baits/proteins that interact with Asf-1 only a small amount of the CT domain is required and it is argued that the binding is due to sequences C-terminal to the conserved CT region.  They try to pin it on some conserved basic sequences at the end of the CT domain and more C terminal but the experiment is two crude deletions and not a set of careful substitutions.  Although briefly discussed in the discussion I wonder whether the genetic design at the outset could have been improved to only remove the CT amino acids raising the question if the mutant had removed only the CT amino acids would the same effects be observed.  The paper as it stands seems confounded by this.  

Fourth, it is nice to measure Kd s but 2microM seems to be too high to be useful.  What is the concentration of Hairless in a cell?

Round 2

Reviewer 2 Report

I am satisfied with the changes made to the manuscript.